# Asymmetric adhesion of rod-shaped bacteria controls microcolony morphogenesis

Marie-Cécilia Duvernoy[1,2,6], Thierry Mora[1], Maxime Ardré[1,6], Vincent Croquette[1,6], David Bensimon[1,3,6], Catherine Quilliet[2], Jean-Marc Ghigo[4], Martial Balland[2], Christophe Beloin[4], Sigolène Lecuyer[2] & Nicolas Desprat [1,5,6]

Surface colonization underpins microbial ecology on terrestrial environments. Although factors that mediate bacteria–substrate adhesion have been extensively studied, their spatiotemporal dynamics during the establishment of microcolonies remains largely unexplored. Here, we use laser ablation and force microscopy to monitor single-cell adhesion during the course of microcolony formation. We find that adhesion forces of the rod-shaped bacteria *Escherichia coli* and *Pseudomonas aeruginosa* are polar. This asymmetry induces mechanical tension, and drives daughter cell rearrangements, which eventually determine the shape of the microcolonies. Informed by experimental data, we develop a quantitative model of microcolony morphogenesis that enables the prediction of bacterial adhesion strength from simple time-lapse measurements. Our results demonstrate how patterns of surface colonization derive from the spatial distribution of adhesive factors on the cell envelope.

[1] Laboratoire de Physique Statistique, École Normale Supérieure, PSL Research University, Université Paris Diderot Sorbonne Paris-Cité, Sorbonne Université UPMC Univeristé Paris 06, CNRS, 24 rue Lhomond, 75005 Paris, France. [2] LIPHY, Univ. Grenoble Alpes - CNRS, F-38000 Grenoble, France. [3] Department of Chemistry and Biochemistry, UCLA, Los-Angeles CA 90095, USA. [4] Genetics of Biofilms, Institut Pasteur, 25-28 rue du Dr. Roux, 75015 Paris, France. [5] Paris Diderot University, 10 rue Alice Domon et Leonie Duquet, 75013 Paris, France. [6] Institut de biologie de l'Ecole normale supérieure (IBENS), Ecole normale supérieure, CNRS, INSERM, PSL Research University, 46 rue d'Ulm, 75005 Paris, France. Correspondence and requests for materials should be addressed to N.D. (email: desprat@ens.fr)

In natural environments, microorganisms compete for resources, but also for space[1]. On solid surfaces, bacterial communities form organized structures[2,3] that arrange with a wide spectrum of morphologies ranging from simple regular pancake shapes for domesticated strains or wrinkled colonies for natural isolates[4,5] to streamers[6], mushroom-like structures in *Pseudomonas aeruginosa*[7] and fruiting bodies in *Bacillus subtilis*[8] and *Myxococcus xanthus*[9]. The morphology of surface-attached communities depends on external parameters such as flow velocity[6] or carbon sources[10] and adapts to environmental structures like oxygen gradients[11,12]. As a result, spatial heterogeneities emerge inside these dense communities[13,14] and induce patterns of proliferation rate[15] and gene expression[16–18].

Bacteria colonize surfaces at the solid–liquid[7], solid–air[12], or liquid–air[11] interfaces. They are also able to invade very confined spaces, for instance porous media[19], biological tissues[20], or the surface of implanted medical devices[21]. During surface colonization, bacterial adhesion plays a central role in maintaining the physical contact between the bacteria and the substrate[22]. Therefore, the envelope of Gram-negative bacteria displays an arsenal of macromolecules and appendages[23], spanning from surface adhesins[24] to polysaccharides[25], which mediate adhesion to biotic or abiotic surfaces[26,27].

Surface-attached communities are seeded by single adhering bacteria, which after few division cycles form microcolonies. The physics of microcolony morphogenesis involves complex mechanical couplings between cell elongation forces[28], adhesion, friction[29], and also cell rearrangements and steric interactions[30]. Initially bacteria proliferate within a monolayer, the extent of which depends on the level of confinement: microcolonies quickly become 3D after very few divisions when growing at solid–liquid interfaces[31], whereas they form wide monolayers when confined between glass and agarose[32,33]. During monolayer expansion, proliferating bacteria adhere to the surface. However, the turgor pressure that builds up within bacteria during cell elongation is quite large[34,35] compared to forces that bacteria–substrate adhesions can sustain[36–38]. In consequence, when bacteria are crowded, they push on each other and are susceptible to detach neighboring cells and even rupture their own adhesions. Hence, it is not clear how cell adhesion and cell elongation forces dynamically and spatially coordinate in order to shape surface-attached communities.

Here we set out to understand how a microcolony can grow into a reproducible shape under different conditions and how bacteria maintain physical contacts with surface while proliferating on it. For this purpose, we measure the spatial dynamics of adhesion at both single cell and microcolony levels in surface-attached communities of rod-shaped bacteria. We show that adhesion to the substrate is stronger at the old pole of individual bacteria, creating adhesion foci at the scale of the microcolony. We further develop a quantitative model of microcolony morphogenesis that captures the mechanical rule explaining the transition from a monolayer of bacteria to a multilayered microcolony. Our results highlight how the distribution and the strength of adhesions on the bacterial surface shape the patterns of surface colonization.

## Results

**Substrate adhesion constrains microcolony morphogenesis.** We started by investigating how elongation and adhesion combine to reproducibly shape microcolonies of *Escherichia coli*, a ubiquitous colonizer involved in nosocomial diseases. First, we examined if patterns of growth within the microcolony could contribute to its shape. We tracked individual bacteria in microcolonies growing between glass and agarose (Fig. 1a, Supplementary Fig. 1a and

Supplementary Movie 1), and showed that bacteria elongate at the same rate regardless of their position within the microcolony (Supplementary Fig. 1b). As a result, bacteria are pushed outward during microcolony expansion (Supplementary Fig. 1c, d), and oldest cells[39] remain at the periphery (Supplementary Fig. 2), where cells experience larger displacements (Supplementary Fig. 1e, f). Although cell elongation uniformly drives the expansion of the microcolony, steric interactions between rod-like bacteria contribute to microcolony shape anisotropy, since neighboring bacteria tend to align (Supplementary Fig. 3).

We then determined the contribution of bacteria–substrate adhesion to colony morphogenesis by comparing the wild-type strain (WT) with mutants of adhesion. We studied the effect of the absence of proteinaceous surface appendages ($\Delta_4adh$: absence of flagella, Ag43, type 1 fimbriae and curli) and exopolysaccharides ($\Delta_4pol$: absence of Yjb polysaccharide, cellulose, poly N-acetylglucosamine or PGA and colanic acid). We quantified the shape of microcolonies by measuring their aspect ratio (Fig. 1b) and showed that reduced levels of cell–substrate adhesion generate more elongated microcolonies (Fig. 1c and Supplementary Movie 2). Past a certain size, a second layer of cells (Fig. 1b, d) forms at the center of the microcolony[32]. The transition from one monolayer to multilayers has been shown to depend on the rigidity of the gel[33] and simulations have suggested that adhesion must be involved[40]. For a given rigidity, we showed that this transition occurs at larger microcolony size in adhesion mutants (Fig. 1e). Therefore, the level of bacterial adhesion influences both the shape of the microcolony and the transition from two-dimensional (2D) to three-dimensional (3D) growth. However, the size at second layer formation does not correlate with microcolony shape. To gain an understanding of microcolony morphogenesis, we performed single cell experiments.

**Asymmetric adhesion at cellular level induces mechanical tensions.** Heterogeneity in bacterial adhesion at the single cell level could also influence microcolony shape. Since cell elongation and steric repulsion cause collective rearrangements in the microcolony, that may screen the influence of adhesion, we monitored isolated bacteria. For the first cell cycle, we measured the movement of the center of mass (CM), $\Delta X$ and cell elongation, $\Delta L$, in order to quantify the asymmetry of adhesion $A_{cell}$ through the relationship $\Delta X(t) = A_{cell}\Delta L(t)$ (Fig. 2a). Since isolated bacteria elongate symmetrically around their CM[41–45], uniform adhesion must yield a null asymmetry parameter ($A_{cell} = 0$). On the contrary, we noted that the CM moves during cell growth (Supplementary Movie 3), indicating that adhesion is not uniform along the cell envelope (Supplementary Fig. 4a, b). We measured that the level of asymmetry $A_{cell}$ is reduced in adhesion mutants (Fig. 2b). Our results are consistent with previous observations that showed polar localization of several adhesion factors[46–49]. We further evidenced by immunofluorescence that Ag43, one of the deleted factors in $\Delta_4adh$, is localized at one of the two poles in *E. coli* cells (Supplementary Fig. 4c). Noticeably, the degree of asymmetry correlates with microcolony shape. To test if asymmetric adhesion is a particular trait of the enterobacteria *E. coli*, we conducted the same experiments on *P. aeruginosa*, a Gram-negative bacteria which belongs to a different genus. We compared a WT *P. aeruginosa* strain to the cupA1 fimbriae mutant, and found the same trend (Fig. 2b), suggesting that asymmetric polar adhesion could be a general feature of rod-shaped Gram-negative bacteria.

To determine which pole carries most of the adhesion, we tracked successive divisions while preventing steric interactions between daughter cells. To do so, we ablated one of the two daughter cells after each division. We first performed laser

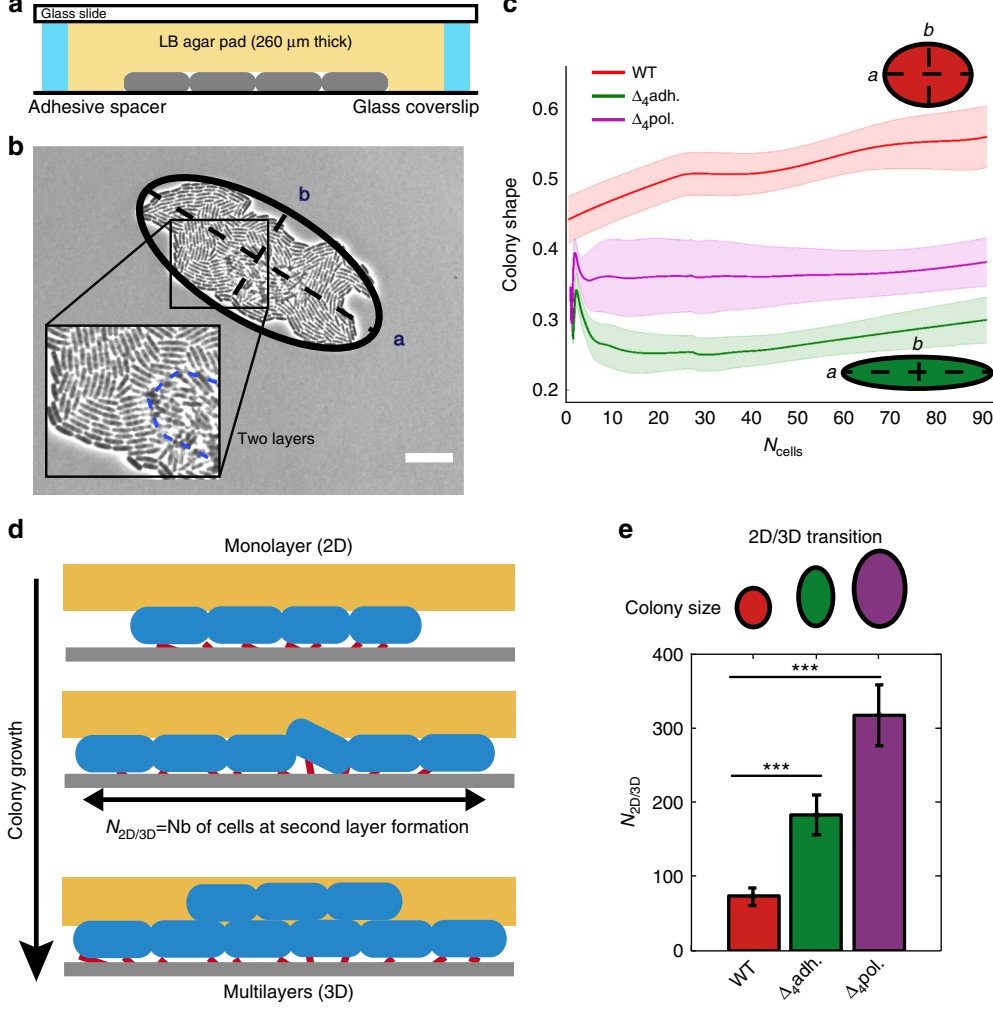

**Fig. 1** Influence of bacterial adhesion on microcolony morphogenesis. **a** Schematic of the culture chamber. Bacteria are confined between an agarose pad and a glass coverslip. **b** Image of a wild type *E. coli* microcolony after the formation of a second layer. The microcolony is fitted with an ellipse with long axis *a* and short axis *b*. Scale bar represents 20 μm. **c** Colony shape measured by the aspect ratio (*b*/*a*) for WT *E. coli* and mutants impaired in adhesive molecules as a function of the number of bacteria in the microcolony. **d** When microcolonies exceed a certain size ($N_{2D/3D}$), a second layer forms on top of the initial one. **e** Microcolony size at the onset of second layer formation. In **c** and **d**, error bars are defined as s.e.m. (WT *E. coli*, red, *N* = 7; $\Delta_4$pol purple, *N* = 5; $\Delta_4$adh green, *N* = 5). Statistical analysis was performed using the unpaired *t*-test (\*\*\**p*-value < 0.001)

ablations on the same side of the pair of daughter cells, so that the same old pole was conserved throughout the experiment (Fig. 2c, Supplementary Movie 4 and Supplementary Fig. 5). After the first division, we were able to identify new and old poles, and orient the cell axis. The positive signed asymmetry $A_{cell}$ shows that adhesion is stronger at the old pole (Fig. 2c). To rule out possible artifacts due to the geometry of the ablation procedure, we performed ablations on alternating sides, so that the old pole was renewed at each generation (Supplementary Movie 4 and Supplementary Fig. 5a). Even in this configuration, $A_{cell}$ remained positive (Fig. 2c). The similar values obtained in the two configurations show that adhesion at the old pole fully matures over one cell cycle. This rapid maturation enables bacteria to re-attach once they have been detached by growing neighbors.

Upon division, the old poles of two daughter bacteria are located on the opposite sides of the cell pair (Fig. 2d). Since bacteria adhere more strongly at their old pole, they tend to elongate toward each other. This situation favors buckling instability[30,50] that triggers rapid reciprocal repositioning of the two daughter bacteria (Supplementary Fig. 4d, Supplementary Movie 5). For WT *E. coli*, we observed that the magnitude of this

reorganization was correlated with the level of asymmetry of the mother cell (Supplementary Fig. 4e). These results illustrate how polar adhesion, coupled with bacterial elongation, can generate mechanical tensions.

**Adhesion is heterogeneous and dynamic during surface colonization**. To further explore the dynamics of substrate adhesion during colony expansion, we monitored the adhesive force patterns in growing *E. coli* and *P. aeruginosa* microcolonies. For time-resolved force microscopy experiments[51,52], bacteria were confined between a rigid agarose gel and a soft polyacrylamide (PAA) gel, in which embedded fluorescent beads served as deformation markers. We measured the substrate deformation field to calculate the adhesive stress generated by mechanical tensions of the growing colony. We found that adhesive stress is heterogeneous and dynamic (Fig. 3a, Supplementary Movies 6 and 7). However, the global stress $\sigma_{colo}$ remains constant during growth (Supplementary Fig. 6a, b), and the radial average stress is uniform (Supplementary Fig. 6c). We also found that adhesion mutants display lower $\sigma_{colo}$ (Fig. 3b). We then measured the

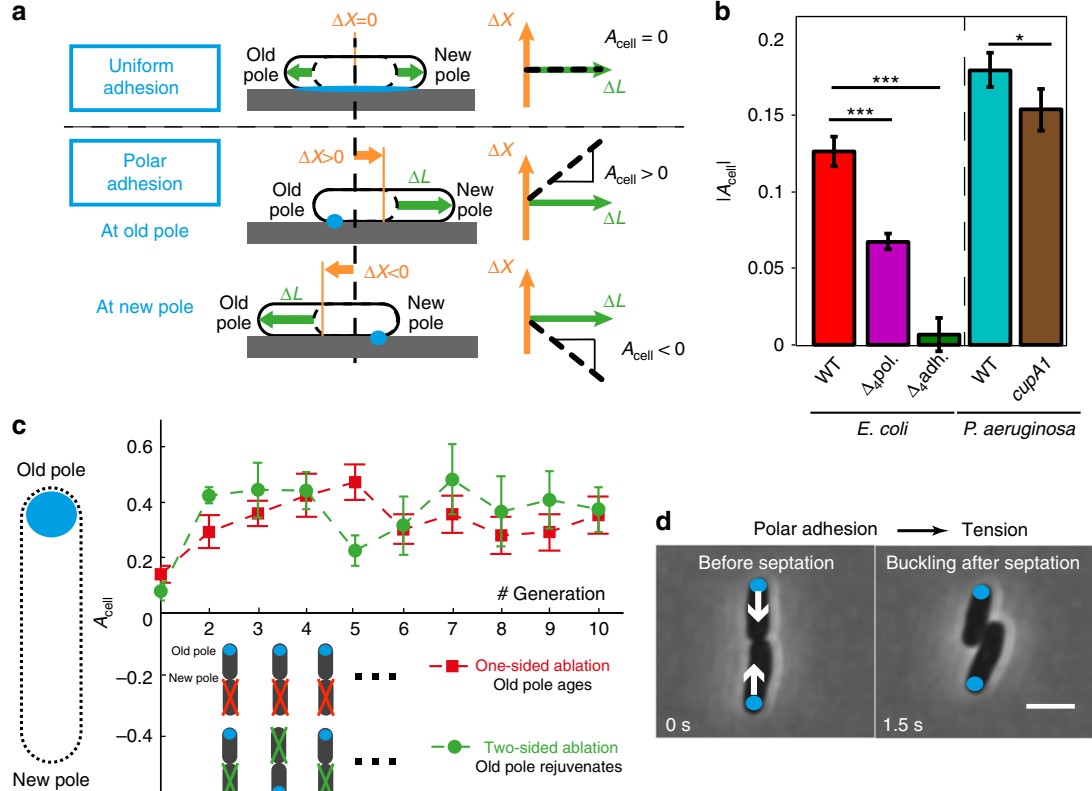

**Fig. 2** Asymmetric adhesion puts daughter cells under tension. **a** Polar adhesion translates into a displacement of the center of mass (CM) of a growing cell. The level of adhesion asymmetry of a cell, $A_{cell}$, is defined according to $\Delta X = A_{cell}\Delta L$, where $\Delta X$ is the displacement of the CM projected along the long axis of the cell and $\Delta L$ is cell elongation. The cell axis is oriented from the old pole toward the new pole. Asymmetry $|A_{cell}|$ may vary from 0 (uniform adhesion) to 0.5 (polar adhesion). **b** Average asymmetry $|\langle A_{cell}\rangle|$ in populations of isolated bacteria for different strains: WT *E. coli* (red, $N = 146$) and different mutants ($\Delta_4 adh$ green, $N = 95$; $\Delta_4 pol$ purple, $N = 159$) adhesion factors; WT *P. aeruginosa* PA14 (cyan, $N = 111$) and fimbriae mutant *cupA1* (brown, $N = 98$). Error bars are defined as s.e.m. Statistical analysis was performed using the unpaired *t*-test of the fitted distribution (***$p$-value < 0.001; *$p$-value < 0.05; $p$-value(PA14 vs. *cupA1*) = 0.047). **c** Successive laser ablations of one daughter after the division of WT *E. coli* enable assessing the sign of $A_{cell}$. Positive $A_{cell}$ indicates that bacteria adhere to the substrate at their old pole. **d** Images of WT *E. coli* before and after the reorganization that follows septation. Old poles are marked by a blue dot. Scale bar represents 3 μm

maximal local force $F_{max}$ generated on the substrate throughout the growth of the microcolony. $F_{max}$ increases early on, and then saturates. Interestingly, saturation is reached when the microcolony becomes larger than about eight bacteria (Supplementary Fig. 6d), corresponding to one cell being fully surrounded by neighbors[53]. The value of the force plateau, $F_{foci}$, is consistent with adhesion forces measured on single cells by Atomic Force Mircroscopy[37,38,54]. As for $\sigma_{colo}$, $F_{foci}$ is reduced for mutants of adhesion (Fig. 3c). Together, our results show that bacteria interact with the substrate in discrete adhesion foci, with local stress fluctuations bounded by the strength of adhesion of individual bacteria.

**Tension at adhesion foci sets second layer formation**. A possible explanation to the dynamics of adhesion foci could be that during microcolony expansion, elongation forces of surrounding bacteria break the polar adhesive bonds. To test this hypothesis, we tracked individual poles in the colony, and found that local adhesive forces indeed drop when a pole undergoes sudden displacement (Fig. 3e and Supplementary Fig. 7a). In agreement with experiments on individual bacteria, we observed that adhesive forces are stronger at old poles throughout colony growth (Supplementary Fig. 7b). Because most bacteria are constantly pushed by others, not all poles can maintain their adhesion to the substrate (Supplementary Fig. 7c, d). While rupture of individual adhesive bonds enables 2D expansion of the microcolony,

formation of a second layer at the center of the microcolony (Fig. 1d)[32,33] suggests that bacterial elongation cannot overcome the cumulative adhesion of surrounding bacteria above a critical colony size. Indeed, our experimental measurements show that the strength of adhesion to the substrate, $F_{foci}$, directly sets the number of bacteria in the microcolony at the onset of second layer formation (Fig. 3d). To test the influence of environmental parameters, we first lower the temperature from 34 to 28 °C. In WT *E. coli* strain, lowering the temperature reduces asymmetry $A_{cell}$, average stress $\sigma_{colo}$, and force at adhesion foci $F_{foci}$ (Supplementary Fig. 8). However, the *E. coli* mutant strain *ompR234*, which overexpresses the curli adhesin at 28 °C[55], displays higher asymmetry $A_{cell}$, average stress $\sigma_{colo}$, and adhesion force at foci $F_{foci}$ (Supplementary Fig. 9). Then, we studied if the composition of the medium can also influence the adhesive properties. Since we have previously shown that *P. aeruginosa* forms a second layer at a smaller microcolony size under low iron conditions[56], we performed force microscopy experiments and indeed confirmed that adhesion foci reinforce under low iron conditions (Supplementary Fig. 10).

**Physical model for surface colonization by rod-shaped bacteria**. Numerical simulations of microcolony morphogenesis are either performed on lattices using coarse-grained models[30,57] or involve objects with steric interactions[33,40,58–60]. In addition to existing steric models, we described adhesive links using real polymer

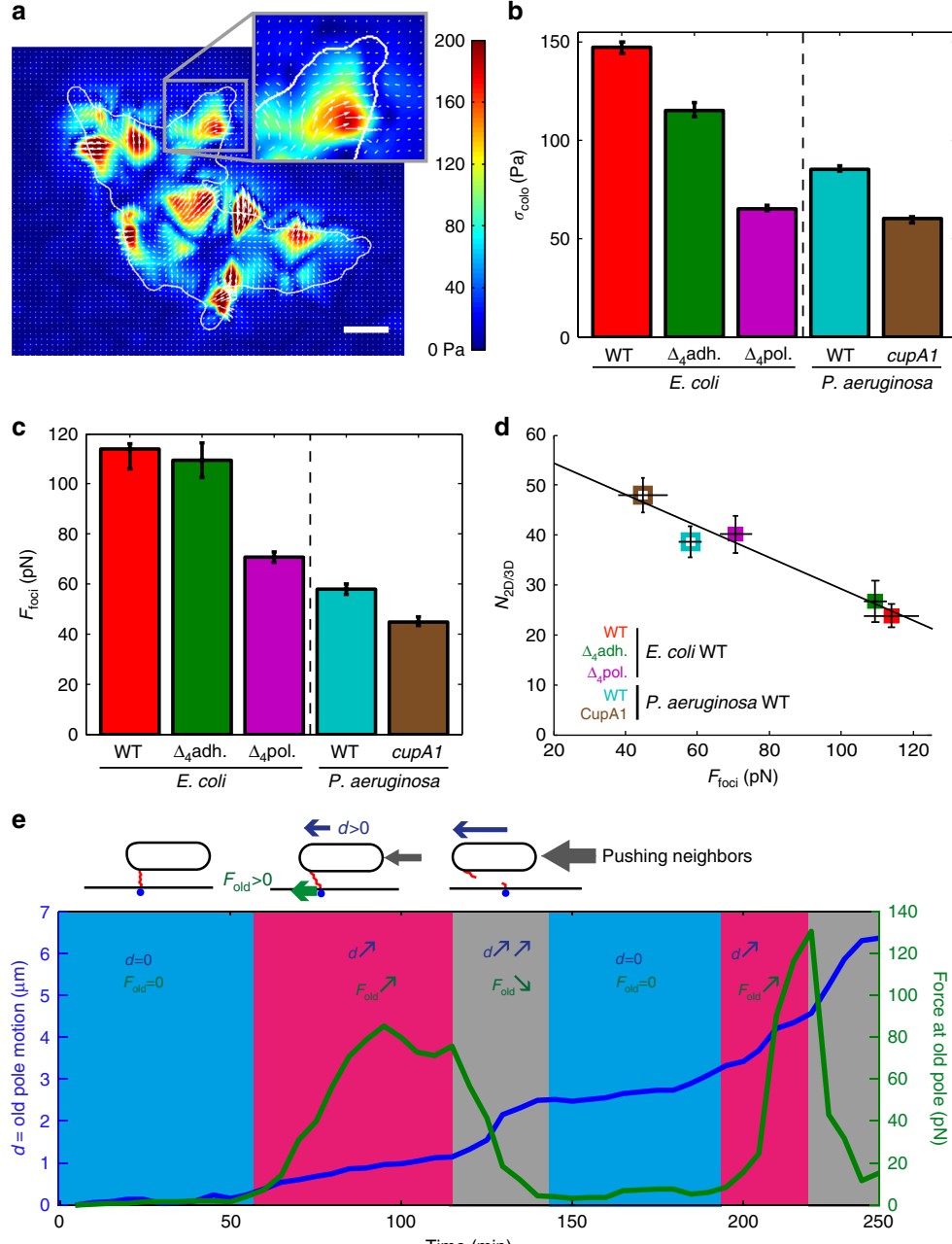

**Fig. 3** Cell–substrate adhesion dynamics within the microcolony. **a** Typical stress field that bacteria exert on the substrate in a WT *E. coli* microcolony (white contour), with a magnified region for better visibility (inset). White arrows show the direction of the forces. Scale bar represents 3 μm. **b** Global stress $\sigma_{colo}$ generated by adhesion forces in WT *E. coli* (red, $N = 12$) and mutants ($\Delta_4 adh$, green, $N = 19$; $\Delta_4 pol$, purple, $N = 48$); WT *P. aeruginosa* PA14 (cyan, $N = 16$), and fimbriae mutant *cupA1* (brown, $N = 14$). **c** Maximal local force measured at adhesion foci, $F_{foci}$. **d** Number of bacteria inside the microcolony at the onset of second layer formation ($N_{2D/3D}$) as a function of the force in adhesion foci ($F_{foci}$). **e** Dynamics of adhesive force (green) and displacement (blue) at an individual pole. If the pole moves slightly, very little force is exerted on the substrate (blue phase). The force increases when the pole moves (pink phase). The force then drops following abrupt displacement (gray phase)

physics and tracked their spatiotemporal dynamics across generations. Adhesive elastic links form at a fixed rate at both poles of bacteria. Since new poles are free of adhesive molecules right after septation, asymmetric adhesion is a natural consequence of cell division. Once exposed on the cell envelope, adhesive links interact with the substrate and stretch while bacteria elongate. They break when the force they experience becomes larger than the rupture force $F_{link}$. The force at adhesion foci $F_{foci}$ is directly set by $F_{link}$ (Supplementary Fig. 11a). For a rupture force $F_{link}$ of 4.25 pN corresponding to the experimental value $F_{foci} = 115$ pN

measured for WT *E. coli*, our simulation reproduces microcolony shape in good agreement with the experimental results (Fig. 4a), and mimics the dynamics and heterogeneity of adhesive stress (Supplementary Movies 8 and 9). In line with our experiments, weaker adhesive links generate more elongated microcolonies (Supplementary Fig. 11b–d).

Bacteria growing in close contact on a surface exert repulsive and adhesive forces. During monolayer expansion, repulsive and adhesive energies rise with the number of cells in the microcolony. Bacteria continue to proliferate within a monolayer

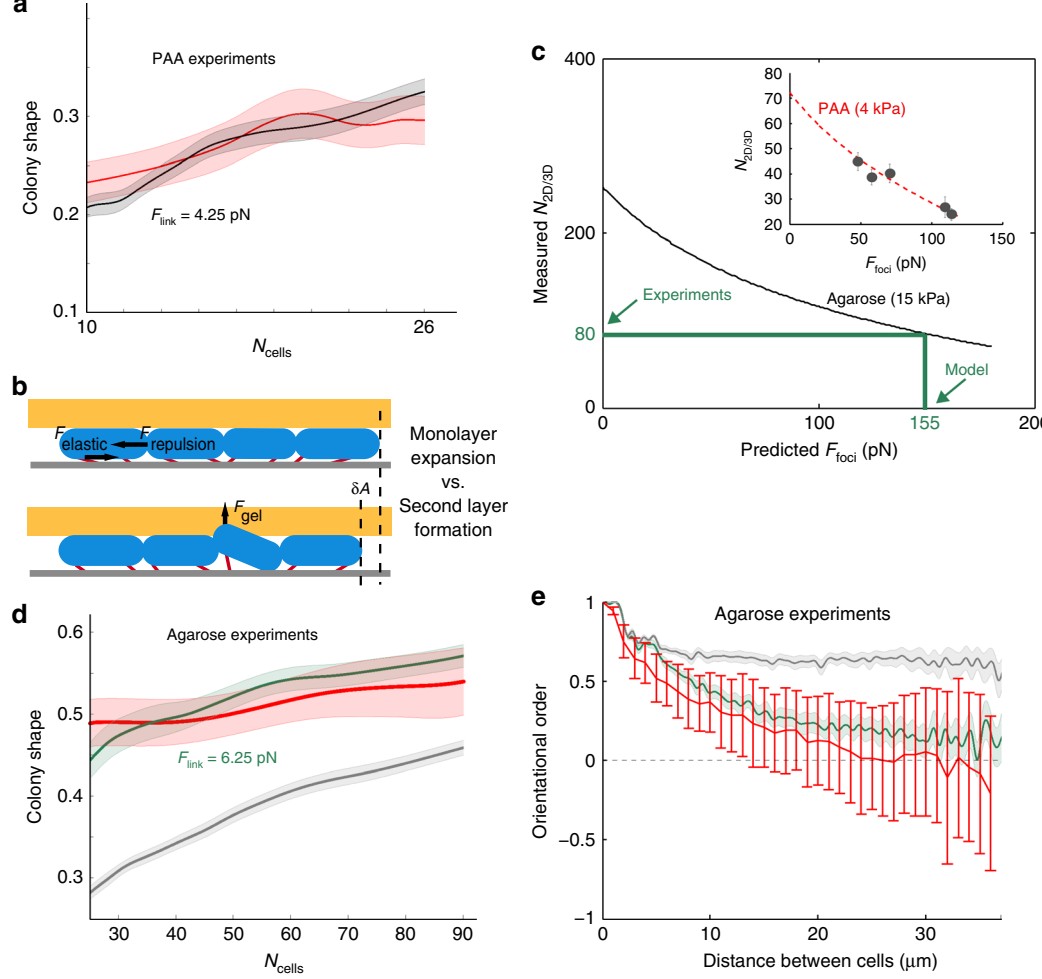

**Fig. 4** Simulation of microcolony morphogenesis and force prediction. **a** Measured aspect ratio of WT *E. coli* microcolonies grown in force microscopy experiments (red, $N = 26$) compared to simulations (black: $N = 10$, $F_{link} = 4.25$ pN). **b** When the microcolony expands in the plane (upper panel), both the elastic energy of the links ($E_{adh}$) and the repulsive energy of steric interactions ($E_{rep}$) increase. In contrast, when a bacterium goes into 3D and initiates second layer formation, it releases its repulsive energy and the microcolony saves a small area $\delta A$ in surface expansion . Yet, the bacterium must pay a cost in order to deform the gel and to extend its elastic links in the vertical direction. **c** Scaling between the size of the microcolony at second layer formation ($N_{2D/3D}$) and the force in adhesion foci ($F_{foci}$) for an agarose gel (15 kPa, black line). Thanks to the relation between $N_{2D/3D}$ and $F_{foci}$ (see Eq. (20) in Methods), we inferred the force at adhesion foci $F_{foci}$ in glass–agarose experiments from the number of bacteria at the onset of the second layer (Fig. 1e). In the inset, comparison between the theory for a soft PAA gel (4 kPa, red dashed line) and experimental values reported in Fig. 3d (dark gray dots). **d** Comparison of colony shape between experimental data of WT *E. coli* microcolonies (red) in glass–agarose experiments (Fig. 1a) and simulations (polar, green; uniform, gray). **e** Comparison of the orientational order in microcolonies between experimental data on WT *E.coli* (red) and simulation (polar, green; uniform, gray). In **d** and **e**, the simulations are defined as polar when adhesive links form only at poles, and uniform when they form homogeneously along the bacterium

until the energy cost that a single bacterium must pay to deform the gel above becomes lower than increasing the mechanical energy within the monolayer. To understand the transition from 2D expansion to 3D growth, we investigated how elastic energy stored in adhesive links $E_{adh}$ and repulsive energy between bacteria $E_{rep}$ scale with the size of the microcolony $N_{cells}$ and the force at adhesion foci $F_{foci}$. We found that $E_{adh} = \alpha F_{foci}^2 N_{cells}$ and $E_{rep} = (\beta_0 + \beta_1 F_{foci}) N_{cells}^2$ (Supplementary Fig. 11e, f). Through these scalings, we derived an analytical expression (see Methods) for the size of the microcolony at the 2D–3D transition, $N_{2D/3D}(E, F_{foci})$ that depends on the Young modulus $E$ of the soft gel and the force at adhesion foci $F_{foci}$ (Eq. (20) in Supplementary Materials, Fig. 4b and Supplementary Fig. 12). For the Young modulus $E$ of PAA gel, this analytical expression describes the 2D–3D transition measured in force microscopy experiments (Fig. 4c inset).

Using the Young modulus $E$ of agarose gel (Fig. 4c), we tried to infer the adhesion forces in glass–agarose experiments where they cannot be directly measured. We thus inverted the expression of $N_{2D/3D}(E, F_{foci})$ and deduced the force at adhesion foci $F_{foci}$ corresponding to the experimental value of $N_{2D/3D}$ measured at the onset of second layer formation in glass–agarose experiments (Fig. 4c). To test the validity of our model, we performed simulations with $F_{link}$ set to the value corresponding to the predicted $F_{foci}$. The simulations were able to reproduce both the organization and the shape of real microcolonies growing between glass and agarose (Fig. 4d, e).

In order to address the significance of polar adhesion, we further performed simulations using the same value for $F_{link}$ but in which adhesive links were homogeneously distributed along the cell envelope. For the same set of parameters, we showed that uniform adhesion generates more elongated microcolonies

(Fig. 4d) because bacteria are aligned over larger length scales (Fig. 4e). Besides providing an estimate of adhesion forces from the size of the microcolony at second layer formation, our model thus demonstrates that the subcellular distribution of adhesive links influences the shape of the microcolony.

## Discussion

We show that both the forces at adhesion foci and their asymmetric distribution on the cell envelope contribute to microcolony shape. Thus, decreasing either adhesion forces or the asymmetry in their distribution generates more elongated microcolonies. The force at adhesion foci depends on the nature and the number of adhesive links engaged in the interaction with the surface. WT *E. coli* exert weaker forces in PAA–agarose experiments than in glass–agarose experiments, where forces are computed from $N_{2D/3D}$. Accordingly they form more elongated microcolonies in PAA–agarose experiments. On both substrates, $\Delta_4pol$ exert weaker forces than $\Delta_4adh$. However, $\Delta_4adh$, which displays a higher value of $F_{foci}$ but a lower asymmetry $A_{cell}$ than $\Delta_4pol$ in glass–agarose experiments, forms more elongated microcolonies. Hence, asymmetry appears to be the dominant factor that sets the microcolony shape.

Our results illustrate how spatial dynamics of adhesion on the cell envelope controls the shape of bacterial communities, and how these levels of organization are coupled. Surface adhesion is highly dynamic, allowing bacteria to maintain contact with the surface as the microcolony expands. The shape of the microcolonies depends on the strength of adhesion, but first and foremost on its subcellular localization. Higher asymmetry generates more circular microcolonies. The similar behavior observed for two rod-shaped species belonging to very different genera suggests that asymmetric polar adhesion is broadly relevant. Asymmetric adhesion promotes the formation of circular microcolonies where differentiation strategies can emerge, as the inner and outer bacteria experience different local environments[15,61].

In addition to the spatial organization of adhesive molecules on the cell wall, cell shape also contributes to establish the success[62] and the patterns of surface colonization[63]. Bacterial microcolonies exhibit a wide spectrum of shapes that vary from filaments to circular structures. Although the shape of rod-like bacteria constrains cell orientation within microcolonies, asymmetric adhesion introduces orientational disorder, which enables microcolonies to become rounder in shape. Since orientational disorder introduces gaps within the microcolony, rod-shaped bacteria with asymmetric adhesion can colonize larger territories using less biomass.

Finally, since the spatial distribution of adhesive molecules on the cell envelope can tune the balance between cell-to-cell and cell–environment interactions at the scale of the community, one can speculate that bacteria could actively regulate it in order to promote different patterns of surface colonization in diverse ecological contexts. Yet, understanding the mechanisms that generate the asymmetry awaits resolved spatiotemporal descriptions of the factors involved at the cell envelope[64].

## Methods

**Bacterial strains**. Strains and primers are described in Supplementary Tables 1 and 2. All *E. coli* strains were derived from strain MG1655 (*E. coli* genetic stock center CGSC#6300) and were constructed using the λ red linear DNA gene inactivation method using the pKOBEG plasmid[65,66] followed by P1vir transduction into a fresh *E. coli* background or alternatively by P1vir transduction of a previously constructed and characterized mutation or insertion. We targeted the four major cell surface appendages of *E. coli*, i.e., flagella, type 1 fimbriae, Ag43, and curli, and the four known exopolysaccharides of *E. coli*, i.e., Yjb, cellulose, PGA, and colanic acid. For *P. aeruginosa*, the reference strain PA14 was used, as well as its fimbriae-deficient mutant cupA1::MrT7, obtained from the PA14 transposon insertion mutant library (Ausubel lab[67]).

**Microscopy and image analysis**. Strains were inoculated in lysogeny broth (LB) from glycerol stocks and shaken overnight at 37 °C. The next day, cultures were diluted and seeded on a gel pad (1% agarose in LB). The preparation was sealed on a glass coverslip with double-sided tape (Gene Frame, Fischer Scientific). A duct was previously cut through the center of the pad to allow for oxygen diffusion into the gel. Temperature was maintained at 34 or 28 °C using a custom-made temperature controller[68]. Bacteria were imaged on a custom microscope using a 100×/ NA 1.4 objective lens (Apo-ph3, Olympus) and an Orca-Flash4.0 CMOS camera (Hamamatsu). Image acquisition and microscope control were actuated with a LabView interface (National Instruments). Segmentation[69] and cell lineage were computed using a MatLab code developed in the Elowitz lab (Caltech)[70]. For microcolony analysis, cultures were diluted $10^4$ times in order to obtain a single bacterium in the field of view. Typically, we monitored four different locations; images were taken every 3 min in correlation mode[56].

**Morphological measurements**. Experiments were performed using a confocal microscope (Leica, SP8). The aspect ratio is defined as $\frac{b}{a}$, where $a$ and $b$ are, respectively, the large and small characteristic sizes of the microcolony. We measured $a$ and $b$ by fitting the mask of the microcolony with an ellipse having the same normalized second central moments. The aspect ratio accounts for the anisotropy of the shape. It is close to zero for a linear chain of bacteria and close to one for a circular microcolony. All measurements listed in Supplementary Table 3 are performed before the appearance of the second layer, which is detected manually in the time-lapse sequence.

**Force microscopy**. For experiments carried out at 34 °C, bacteria were grown overnight in LB (37 °C, 200 rpm), diluted 100-fold in 5 mL of fresh LB and grown again in the same conditions for 2 h prior to experiments. Then, the fresh culture was diluted 500 times and seeded on a 2% LB-agarose pad. To promote expression of curli fibers, some experiments were also carried out at 28 °C and bacteria were taken from cultures at saturation. In that case, a $10^4$ dilution from an overnight culture at 28 °C was directly seeded on the 2% LB-agarose pad. Under both conditions, a 4-kPa PAA gel with 200 nm fluorescent beads (FC02F, Bangs Laboratories) embedded below the surface was prepared[51]. The PAA gel bound to a glass coverslip was sealed onto the agarose pad with a double-sided tape. Imaging was performed through the glass coverslip and the PAA gel with an inverted Olympus IX81 microscope. Fluorescence excitation was achieved with a mercury vapor light source (EXFO X-Cite 120Q). Beads were imaged through a 100×/NA 1.35 objective lens (Apo-ph3, Olympus) and an Orca-R² CCD camera (Hamamatsu) with a YFP filter set (Semrock). The microscope, camera, and stage were actuated with a LabView interface (National Instruments). Bacteria were imaged using phase microscopy. Force calculations were performed as previously described[51,71]. A Fourier transform traction cytometry (FTTC) algorithm, with 0-order regularization, was used to calculate the stress map from the substrate deformation, measured via the displacements of fluorescent beads embedded in the gel. After correction for experimental drift, the fluorescent beads were tracked to obtain a displacement field with high spatial resolution. The first frame of the movie, taken after seeding the sample with bacteria, was taken as the reference for non-deformed gel. The displacement field was measured by a combination of particle imaging velocimetry (PIV) and single particle tracking (SPT). PIV was used to take a first measurement of the displacement field induced by the mechanical interactions between the microcolony and the micro-environment. The obtained displacements were then applied to the reference bead image obtained for non-deformed gel. Relative displacement between this PIV-corrected image and the deformed image was then analyzed using SPT to measure residual displacement with subpixel accuracy. The final displacement field was interpolated on a lattice of characteristic size 510 nm. Stress reconstruction was conducted with the assumption that the substrate was a linear elastic half-space medium. We set the regularization parameter to $10^{-9}$. To estimate the noise in stress reconstruction, we compared the average stress outside the colony where no forces are physically exerted, to the average stress beneath the microcolony (Supplementary Fig. 6a). We derived the force $F_{colo}$ exerted by the microcolony on the substrate by integrating the mechanical stress over the surface covered by the microcolony. We then quantified the average stress $\sigma_{colo}$ beneath the microcolony by fitting the linear relation between $F_{colo}$ and the change in microcolony area. The maximal force was then simply obtained by multiplying the maximal value of the stress on the grid by the lattice elementary size (510 nm × 510 nm). To measure the asymmetry in force at the scale of the microcolony, we compared the maximal forces at new and old poles, rather than the average forces, because most of the poles slide on the substrate. Asymmetry of the microcolony was thus defined as follows: $A_{colo} = \frac{\max\{F_{oldpole}\}_{cells} - \max\{F_{newpole}\}_{cells}}{F_{foci}}$.

**Single cell assays**. Asymmetric adhesion assays: Overnight cultures were diluted $10^2$ times in order to obtain on average of 150 bacteria over 10 different fields of view; images were taken every 3 min in phase contrast. Because the two poles of a bacterium are not equivalent in terms of their history[39], we projected the

displacement of the CM $\Delta X$ along the cell axis oriented toward the pole formed after the last division, i.e., the new pole. Since we cannot know pole history until a division has occurred, we measured the absolute value of the parameter of asymmetry $|A_{cell}|$ in a population of isolated cells. Then, we quantified the absolute value of the average $|\langle A_{cell} \rangle|$ by fitting the cumulative distribution of $|A_{cell}|$ with a folded normal distribution (Supplementary Fig. 4b).

Single cell ablation: Ablations were performed using a UV pulsed laser (Explorer 349 nm, Spectra Physics). A train of 30 impulsions at 1 kHz was sent, each delivering to the sample a power density of about $35\,kW.\mu m^{-2}$. Using a custom algorithm on correlation images, live image analysis enabled automatically positioning the laser spot on a chosen bacterium by moving the stage with an $X,Y$ accuracy of 40 nm (Thorlabs, MLS203). In the $Z$ direction, the bacterium is placed at the resolution of our autofocus, i.e., 200 nm. We used a laser with a short wavelength in order to minimize the volume of the focal spot, so that its extension is not larger than the cell width. These experiments were carried out on wild type *E. coli*, the larger size of which, compared to *P. aeruginosa*, enables successive ablations without perturbing the remaining cell (Supplementary Fig. 5b). Since we could not distinguish poles until one division had occurred, we computed the absolute value $|A_{cell}|$ for the first generation.

Reorganization after division: Overnight cultures were diluted $10^3$ times in order to obtain on average of 2–3 bacteria in each field of view; phase contrast images were taken in phase contrast every 30 s before septum formation and every second once the septum was visible.

Immunostaining: Bacteria were grown to OD 0.2 and anti-ag43 was used at a dilution of 1:10,000. Immunostaining was performed in 1.5 mL Eppendorfs. Bacteria were then seeded between an agarose gel and a glass coverslip before image acquisition.

**Model for microcolony morphogenesis.** Bacteria are modeled as spherocylinders that elongate exponentially at rate $g$, $\dot{d}_c = gd_c$, where $d_c$ is the cell length. They are allowed to divide at a constant rate $\alpha$ once they have reached size $d_L$. Division is forced if bacterial length exceeds 30% of $d_L$. The dynamics of bacterial arrangement in the microcolony is driven by the two following effects: (i) cell–cell interactions are modeled with a Yukawa-like potential; and (ii) cell–substrate adhesion is modeled by punctual elastic links that detach above a critical force. Adhesive links are created at the two poles at the same rate. In our model, asymmetric adhesion is a consequence of cell division that gives birth to new poles free of adhesive links. Once they detach, adhesive bonds are lost.

The interaction force between bacteria is derived from a Yukawa potential. For simplicity, each bacterium is modeled as 6 adjacent balls ($b = 1, \ldots, 6$) equally distributed along its length:

$$F^{b,b'}_{cell_{i,j}}(r_{bb'}) = \begin{cases} k_{rep}(r_0 - r_{bb'}) & ; \quad r_{bb'} < r_0 \\ -V_0 \left( \frac{1}{r_{bb'}} + \frac{1}{r_1} \right) \frac{1}{r_{bb'}} \exp\left[ -\left( \frac{r_{bb'} - r_0}{r_1} \right) \right] & ; \quad r_{bb'} \geq r_0 \end{cases} \quad (1)$$

$F^{bb'}_{cell_{i,j}}$ is the interaction force between two adjacent balls that belong to two distinct bacteria ($i, j$) and distant from $r_{bb'}$. $k_{rep}$ is the elastic constant for cell–cell repulsion. The distance of repulsion $r_0$ sets the cell width. $V_0$ sets the potential depth and $r_1$ sets the range of attraction.

Elastic links form at rate $k_{on}$ and their density saturates at $n_l$ links per ball. In the polar case, they randomly appear at either pole in a disk of diameter $r_0$. In the uniform case, they randomly appear all along the spherocylinder. Like polymers, adhesins or polysaccharides elasticity is described by the worm-like-chain model[72]. The force applied on an individual link $l$ in a given ball $b$ is expressed as:

$$F^{b,l}_{adh}(L) = -\frac{kT}{L_p} \left( \frac{L}{L_0} + \frac{1}{4} \left( 1 - \frac{L}{L_0} \right)^{-2} - \frac{1}{4} \right), \quad (2)$$

where $L_0$ and $L_p$ are, respectively, the total and persistence lengths of the polymer. $L$ is the link extension.

The links detach at a rate that depends on the tension exerted on them:

$$r_{detach}\left( F^{b,l}_{adh} \right) = k_{off} \times \begin{cases} 1 + \operatorname{arctanh}\left( \frac{F^{b,l}_{adh}}{F_{link}} \right) & ; \quad F^{b,l}_{adh} \leq F_{link} \\ \infty & ; \quad F^{b,l}_{adh} > F_{link} \end{cases} \quad (3)$$

The threshold in force, $F_{link}$, is a parameter used to vary the strength of adhesion.

We performed overdamped molecular dynamics simulations to model the motion of bacteria:

$$\nu_t \frac{d\vec{X}_i}{dt} = \sum_{b=1}^{6} \left( \sum_{j \in V(i)} \sum_{b'=1}^{6} \vec{F}^{b,b'}_{cell_{i,j}} + \sum_l \vec{F}^{b,l}_{adh} \right), \quad (4)$$

$$\nu_r \frac{d\theta_i}{dt} \vec{e}_z = \sum_{b=1}^{6} \left( \sum_{j \in V(i)} \sum_{b'=1}^{6} \vec{r}_{bb'} \times \vec{F}^{b,b'}_{cell_{i,j}} + \sum_l \vec{r}_{bl} \times \vec{F}^{b,l}_{adh} \right), \quad (5)$$

where $i,j$ are the indices of the cells; $V(i)$ designates the neighboring cells of $i$; $b,b'$ and $l$ are respectively the indices of the balls that constitute cells and the adhesive links between the cell and the substrate; $\nu_t$ and $\nu_r$ are translational and rotational friction coefficients, respectively, for cylinders in a viscous fluid of viscosity $\eta$[73]:

$$\nu_t = \frac{3\pi\eta d_c}{\ln(p) + C_t(p)}, \quad (6)$$

$$\nu_r = \frac{\pi\eta d_c^3}{3(\ln(p) + C_r(p))}, \quad (7)$$

where $p = \frac{d_c}{r_0}$ is the aspect ratio of bacteria. $C_t$ and $C_r$ are given by:

$$C_t = 0.312 + \frac{0.565}{p} - \frac{0.1}{p^2}, \quad (8)$$

$$C_r = -0.662 + \frac{0.917}{p} - \frac{0.05}{p^2}. \quad (9)$$

When bacteria elongate, adhesive links are extended and surrounding bacteria are pushed. As a result, elastic and repulsive energies increase during the 2D expansion of the microcolony. We compute elastic and repulsive energies as follows:

$$E^i_{adh} = \sum_{b=1}^{6} \left( \sum_l \int_0^{L_{link}} F^{b,l}_{adh}(L) dL \right), \quad (10)$$

$$E^i_{rep} = \frac{1}{2} \sum_{b=1}^{6} \left( \sum_{j \in V(i)} \sum_{b'=1}^{6} k_{rep} H(r_0 - r_{bb'})(r_0 - r_{bb'})^2 \right), \quad (11)$$

where, $H$ is the Heaviside function.

We performed simulations at different values of $F_{link}$ and deduced the scalings for adhesion and repulsion energies in the microcolony as a function of microcolony area, $A$, and force at adhesion foci, $F_{foci}$ (Supplementary Fig. 11e, f):

$$E_{adh}(A) = \alpha F_{foci}^2 A, \quad (12)$$

$$E_{rep}(A) = \left( \beta_0 + \beta_1 F_{foci} \right) A^2. \quad (13)$$

As a result, the energy of the microcolony scales as follows as a function of its area $A$:

$$E_{colo}(A) = E_{adh}(A) + E_{rep}(A) = \alpha F_{foci}^2 A + \left( \beta_0 + \beta_1 F_{foci} \right) A^2. \quad (14)$$

Bacteria proliferate in 2D until it is energetically less favorable for the microcolony to exclusively raise the energy in the plane rather than paying the cost required for a bacterium to deform the gel and go 3D. We compared a situation in which the entire increase in surface area d$A$ remains confined in the monolayer to a situation in which a bacterium elongates in the third dimension. For each situation, the mechanical energy of the microcolony increases by $\Delta_{2D}E_{colo}(A \to A + dA)$ and $\Delta_{3D}E_{colo}(A \to A + dA)$, respectively.

When a bacterium goes 3D, a small surface element $\delta A$ does not contribute anymore to monolayer expansion and the repulsive energy of the bacterium is released. But the bacterium has to pay an energetic cost $E_g$ to deform the gel above and a mechanical work $W$ to extend its adhesive links in the third dimension. $\Delta_{2D}E_{colo}(A \to A + dA)$ and $\Delta_{3D}E_{colo}(A \to A + dA)$ are expressed as follows:

$$\Delta_{2D}E_{colo}(A \to A + dA) = E_{colo}(A + dA) - E_{colo}(A), \quad (15)$$

$$\Delta_{3D}E_{colo}(A \to A + dA) = \Delta_{2D}E_{colo}(A \to A + dA - \delta A) + E_g - E^0_{rep}(A) + W. \quad (16)$$

The average repulsion energy $E^0_{rep}$ per bacterium is simply the total repulsion energy $E_{rep}$ (Eq. (13)) divided by the number of bacteria $N_{cells} = \frac{A}{A_0}$ with $A_0$ being the average area of bacteria after division:

$$E^0_{rep}(A) = \frac{E_{rep}(A)}{N_{cells}} = \left( \beta_0 + \beta_1 F_{foci} \right) A A_0. \quad (17)$$

The work $W$ of the adhesive links is proportional to the rupture force of individual links $F_{link}$, itself proportional to $F_{foci}$ and to the elongation of the links $z$ corresponding to the indentation of the gel. Thus, $W = \gamma z F_{foci}$. Similarly, we estimate that $\delta A = z r_0$.

Knowing the Young modulus $E$ and the Poisson ration $\nu$ of the deformable gel, we used the Hertz model to compute the energy required to deform the soft interface.

$$E_g(z) = \int_0^z \frac{4}{3} \frac{E}{1-\nu^2} \left(\frac{r_0}{2}\right)^{1/2} z'^{3/2} dz' = \frac{8}{15} \frac{E}{1-\nu^2} \left(\frac{r_0}{2}\right)^{1/2} z^{5/2}. \qquad (18)$$

At the 2D–3D transition, the two situations have the same energy. Hence,

$$\Delta_{2D} E_{colo}(A \rightarrow A + dA) = \Delta_{3D} E_{colo}(A \rightarrow A + dA). \qquad (19)$$

By inserting $N_{cells} = \frac{A}{A_0}$ into Eq. (19), we compute the number of bacteria at double layer formation $N_{2D/3D}$:

$$N_{2D/3D}(E, F_{foci}) = \frac{E_g(E) + \gamma z F_{foci} - \alpha F_{foci}^2 z r_0}{A_0(2 z r_0 + A_0)(\beta_0 + \beta_1 F_{foci})}. \qquad (20)$$

Using this equation, we fitted the experimental data obtained by force microscopy with two parameters in order to estimate the values of the critical indentation $z$ and $\gamma$. Values of the parameters used in this study are referenced in Supplementary Table 4.

**Data availability**. The datasets generated during the current study are available from the corresponding author.

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

## Acknowledgements

We thank Paul Rainey, Roberto Kolter, Oskar Hallatscheck, and Jean-Baptiste Boulé for their comments on the manuscript. We thank Irene Wang for her help with force microscopy calculations. We thank Michael Elowitz for Schnitzcell. We thank José Quintas Da-Silva and Carlos Gonzales for design and construction of the mechanical parts. We thank Mathieu Coppey, Manuel Thery, Frederic Lechenault, and Sebastien Moulinet for stimulating discussions. This work was supported by the Agence Nationale pour la Recherche ANR-12-JSV5-0007-01 (to M.B.), ANR-10-LABX-62-IBEID (to C.B. and J.-M.G.), and ANR-11-JSV5-005-01 (to N.D.) and by the Fondation pour la Recherche Médicale grant Equipe FRM DEQ20140329508 (to C.B. and J.-M.G.).

## Author contributions

M.-C.D. performed and analyzed the data from force microscopy, laser ablation, and colony growth experiments. M.A. and M.-C.D. analyzed the cell movements in micro-colonies. N.D. set up the laser ablation experiments and did the asymmetric assays on single cells. C.B. did the immunofluorescence experiments on Ag43. T.M., D.B., and N.D. did the modeling. T.M. and N.D. designed and wrote the code for the simulations.M.-C. D., V.C. and N.D. developed the tools for the microscopy. M.B. designed and coordinated force microscopy experiments. J.-M.G. and C.B. determined the set of E. coli strains and relevant experimental conditions for E. coli. S.L. did likewise for P. aeruginosa. N.D., S.L., and C.Q. co-advised M.-C.D. during her PhD. C.B., J.-M.G., S.L., and N.D. wrote the paper.
