## [Peer Review File(PDF 13564 kb) · Nature Communications]

Reviewers' comments:

Reviewer #1 (Remarks to the Author):

This paper reports on experiments probing the adhesion of bacteria to a surface and its effect on colony formation. By combining cell tracking and force microscopy, and relating them to colony shape, the authors demonstrate that asymmetric polar adhesion (with old poles adhering more strongly) contributes to the loss of orientational order in the colony and thereby leads to rounder colonies. Moreover, they show that the transition from 2d growth to 3d growth (second layer of cells) is also governed by adhesion and show a beautiful relation between the force of adhesion at the polar foci and the number of cells in the colony at the time of the transition to 3d growth. Numerical simulations of colony growth are shown to be in good agreement with the experimental observations.

This is an excellent paper, which I highly enjoyed reading. It is very well written and reports well-designed experiments that provide a new picture of the mechanics of colony growth. I would almost have recommended to accept it in its present form.

However, there is one small issue where I would advise a revision and this is related to the scaling argument at the end of the paper, where the authors propose an analytical formula to estimate forces of adhesion from the 2d-3d transition for other surfaces. I think this argument could be presented in more detail; in its current condensed form, I found it rather inaccessible.

Reviewer #2 (Remarks to the Author):

Summary of key results: The authors utilized time-lapse microscopy, laser ablation, traction force microscopy, and modeling to demonstrate that asymmetries in cellular adhesion forces at the subcell level leads to: 1) asymmetric elongation elongation away from the older cell pole; and 2) a randomization of the orientations of cells growing in the monolayer, leading to more circular microcolonies (depending on the level of asymmetric adhesion). They measure surface forces and build a model that predicts when cells get pushed into a second layer due to the resistance to outward spreading from cell-surface attachments. The model seems to reproduce their experimental results. I feel that this was a well written and careful treatment of a problem that is often observed, but to my knowledge, hasn't been quantified and explained from a physical basis. This paper is interesting because it links an asymmetry from the localization of subcellular attachments, due to cell 'age' and the concurrent accumulation of proteins/adhesins, to phenomena on intermediate length scales.

Novelty: Microbial surface colonization in confined geometries is important both environmentally, where thin gaps are often found in porous materials, and medically, where catheters can create similar gaps between mammalian cells and polymeric materials. I believe that this fundamental work will be of broad interest in microbiology, bioengineering, and biophysics since it describes the physical consequences of biological adhesion factors and sets the stage for analyzing biofilm development. My suggestions, comments and questions follow.

Data & Methodology: I found the methods straightforward in explaining the phenomena observed. However, it would be helpful to the reader if the authors summarized the variables in a supplementary table for easy referencing. Perhaps, something along the lines of Supp. Table 3.

Conclusions: Overall, I feel that the experiments and the interpretations sufficiently support the conclusions.

Specific comments on Figs/Data

Figure 2

i. 2a: the text with the light green color and small font is hard to read.

ii. 2c and Supplemental Movie 4

(1) it seems to me that the old pole is always 'facing outward' toward pristine agar that has not been disturbed by the presence of a cell (in both types of ablation). In this system, the cells are surrounded by and confined to a 2D layer between agar and glass. Once a cell elongates, it can make a furrow in the gel (e.g. see: Gloag et al, Proc. Nat Acad Sci 110 (2013)). Please explain how you can be sure that the new pole is elongates "forward" due to the increased adhesion of the old pole and that it isn't simply moving into the empty space (lower resistance) left behind by the ablated cell.

(2): how have you controlled for inadvertent gel ablation?

(3): does cell debris affect elongation?

Figure 3

i. 3a: the inset should be referenced in the caption and indicated in the image.

ii. 3d: to avoid confusion due to small font and symbol size, perhaps the symbols can be made large and/or solid vs filled...

iii. 3c and lines 116-117: Can the authors comment on the very small difference between WT and d_adh cells with respect to the polysaccharide mutants for E.coli? In Fig2b the d_adh cells elongate symmetrically, which implies low and symmetric adhesiveness. Despite this, the forces that d_adh generate are quite close to WT. This seems counter intuitive.

Figure 4

i. 4b: Do the authors think that the elastic link to the surface extends the full length of the width of cells? In other words, for cells that get pushed into the 3D, does the link to the surface remain?

Supplementary Figures

i. Supp Fig 1: Light yellow colors are very hard to read

ii. Supp. Fig 2b: the label for the error bars is missing

iii. line 305 refers to Suppl Fig 3B, but should be 4B.

iv. Supp Fig 7 caption: there is no "3G"

Clarity, Context:

i. On lines 6-7 in the abstract the authors make a very general statement that all rod shaped bacteria have higher adhesiveness at the old poles. In my opinion, to avoid confusion they should clarify this point by first referencing the bacteria that they use and then generalizing to all real and modeled rod-shaped bacteria with the type of asymmetric adhesiveness that E. coli and P. aeruginosa have.

ii. The introduction is quite short, too general, and a bit unclear to adequately provide context, introduce, motivate, and review the literature sufficiently for the interesting problem they address. For example, on line 16, what does "organized structures" mean? What "environmental conditions" are being referred to? Why does bacterial elongation compete with surface attachment? Please clarify these points.

iii. On lines 207-211 the authors introduce a weak argument for the potential ecological advantages that rod shaped bacteria have over coccoids in surface colonization. There are far too many assumptions built into this statement for it to be meaningful. Too many factors come into play during surface colonization to simply say that asymmetric adhesion confers an advantage. For example, if a coccoidal species had a shorter doubling time or strong cell-to-cell adhesion, one could imagine a similar surface colonization program but with the coccoid have the advantage. The authors should

provide more context and support for this statement.

iv. Minor comments on awkward phrasing, grammar,

a) lines 186-191: The wording in these lines could be reworded to be more clear

b) line 202: i think the authors mean "adhesins"

We are very grateful for the constructive remarks addressed by the reviewers. We answer their questions point-by-point below. We have addressed the different issues, and have made the appropriate changes (highlighted in red) in the manuscript.

Answer to reviewer 1

1. *However, there is one small issue where I would advise a revision and this is related to the scaling argument at the end of the paper, where the authors propose an analytical formula to estimate forces of adhesion from the 2d-3d transition for other surfaces. I think this argument could be presented in more detail; in its current condensed form, I found it rather inaccessible.*

We have modified the section "Physical model for surface colonization by rod-shaped bacteria" in the main text (Lines 190 to 220). We also clarified and detailed each steps of the model in the Methods (Lines 375 to 445).

The reasoning that underlies the modeling is the following. We performed numerical simulations at different adhesion forces, i.e. F_{link} , in order to understand how the mechanical energy stored in the monolayer evolves with the size of the microcolony, and how it depends on the force of adhesion. We used the scaling of the simulated forces with colony size to derive an analytical expression for the size of the microcolony at second layer formation by quantifying when it becomes less costly for a bacterium to deform the gel above rather than continuing to increase the energy stored in the monolayer. We used this analytical expression to predict the force of adhesion in glass-agarose experiments from the size of the microcolony at second layer formation. Finally, we tested the validity of these predictions by comparing experiments and simulations. We showed that for the predicted force, our simulations reproduce the shape and the organization of experimental microcolonies.

Answer to reviewer 2

1. However, it would be helpful to the reader if the authors summarized the variables in a supplementary table for easy referencing. Perhaps, something along the lines of Supp. Table 3.

We have added at the end of the manuscript (Line 642) a table S4 that references the name and the definitions of all measured parameters.

2. Figure 2.

i. 2a: the text with the light green color and small font is hard to read.

We have changed the size and the color of the font in Fig2a.

ii. 2c and Supplemental Movie 4

(1) it seems to me that the old pole is always facing outward toward pristine agar that has not been disturbed by the presence of a cell (in both types of ablation). In this system, the cells are surrounded by and confined to a 2D layer between agar and glass. Once a cell elongates, it can make a furrow in the gel (e.g. see: Gloag et al, Proc. Nat Acad Sci 110 (2013)). Please explain how you can be sure that the new pole elongates forward due to the increased adhesion of the old pole and that it isn't simply moving into the empty space (lower resistance) left behind by the ablated cell.

Figure 1: **Single non motile bacteria do not significantly indent the gel.** (a) Image of wild type *E. coli* cells in between a 1% agarose gel and a coverslip. The image is taken half an hour after deposition. (b) Image after removal of the coverslip and washing with distilled water. The image is taken at the same location as in a). (c) Same image as in b) but with enhanced contrast. (d-f) Same image series than in a-c but for another sample.

In Gloag et al. (PNAS 2013), the authors observe groups of pseudomonas cells that navigate by twitching motility in between a gel and a coverslip. They show that moving bacterial rafts can form furrows in the gel that persist after removing the coverslip and washing the bacteria. Those furrows, which are 200nm deep, are observable with phase contrast microscopy. In order to test whether bacteria

form furrows in our experimental context, we followed the same procedure: removed the coverslip and washed bacteria with distilled water. Previously, the gel is homogeneously plated with a solution containing 10^7 bacteria. mL^{-1} in order to get on average 20 bacteria per field of view over a large area (0.5cmx0.5cm). The sample is screwed on a XY motorized stage in order to insure a good repositioning of the sample. We performed observations 30 min after deposition, which roughly corresponds to the average division time of bacteria, *i.e.* the time during which the gel is deformed before ablation. We imaged bacteria in phase contrast and recorded the X,Y position of the image. Then, we unscrewed the sample, removed the coverslip, washed bacteria, put oil on the sample and re-screwed it to the motorized stage. Our images do not reveal the presence of furrows in the gel that correlate with bacteria (Fig. 1). Since manipulations can induce drift in the sample position, we surveyed the neighboring locations. We did not observe significant stains that can be attributed to bacteria.

In Gloag et al. the authors bring very convincing evidences that twitching microcolonies actually form furrows in the gel. The apparent discrepancy with our observations can stem from the fact that we are looking at non-motile isolated bacteria. Indeed, twitching motility involves pili that exert large forces on the gel when they retract. In addition, they use a slightly different protocol that might explain why their gels and ours are not totally equivalent. They used a 0.8% gellan gum gel (a pseudomonas polysaccharide) that is dry for about 30mins before being covered with a coverslip and stored during few days. In our experiments, we use 1% agarose gel (a seaweed polysaccharide). In order to avoid gel aging, we covered the sample immediately after bacteria are deposited and prepared a fresh sample for each experiments.

(2): *how have you controlled for inadvertent gel ablation?*

If ablations are performed inadvertently within the gel, it gives rise to cavitation. The bacteria in the near vicinity are blasted away and the ablation spot appears clearly in the image. In our experiments, ablations are triggered automatically based on a segmentation routine that identifies the location of bacteria within the image. The laser is fixed and an automated stage moves the center of mass of the bacterium into the laser spot with an X,Y accuracy of 40nm. In the Z direction, the bacterium is placed at the resolution of our autofocus, *i.e.* 200nm. We used a laser with a short wavelength (349 nm) in order to minimize the volume of the focal spot, so that its extension is on the order of the cell height. Our experimental protocol thus insures that only a very small fraction of the laser spot is deposited on the gel.

(3): *does cell debris affect elongation?*

Cell debris can generate steric or chemical constraints on the remaining cell. In supplementary figure 5, we showed that the growth rate is fairly constant for the first 10 generations. This indicates that chemical compounds released by the ablated cells do not significantly perturb the elongation rate of remaining bacteria. After many ablations, debris accumulate and we noticed that they can oppose a mechanical resistance to the movement of the new pole. In this case (typically after 10 generations), the old pole can move more than the new pole.

3. Figure 3

i. 3a: the inset should be referenced in the caption and indicated in the image.

ii. 3d: to avoid confusion due to small font and symbol size, perhaps the symbols can be made large and/or solid vs filled?

We have inserted a reference to the inset in the caption of Fig3a. In Fig3d, we have enlarged all symbols. We chose solid symbols for *E. coli* and open symbols for *P. aeruginosa*.

*iii. 3c and lines 116-117: Can the authors comment on the very small difference between WT and Δ_{adh} cells with respect to the polysaccharide mutants for *E.coli*?*

The difference between the adhesive forces measured for Δ_{adh} and Δ_{pol} can be explained by either the number of molecules involved in the interaction with the surface or their respective rupture force. Although those forces have not been simultaneously measured in *E. coli*, AFM experiments performed

on single bacteria with specific antibodies that probe either adhesins (Forero *et al.*, *PLoS Biol* 2006, Jacquot *et al.*, *J. Bio. Nano.* 2014) or polysaccharides (Beaussart *et al.*, *Nanoscale* 2014), have revealed that those adhesive links share almost the same features with regards to the rupture force. However, adhesins are generally more specific to biotic surface (Chagnot *et al.*, *Cellular Microbiol.* 2012, Korea *et al.*, *Bioessays* 2011) while polysaccharides are known to be less specific and more likely involved in the interactions with abiotic surfaces (Matthysse *et al.*, *App. Environ. Microbiol.* 2008). In addition, while the tip of adhesins mediates surface interaction, polysaccharides may form lateral interactions along the sugar chain. In our experiments, the comparison between the size of microcolonies at second layer formation indicates that adhesins contribute more on agarose (biotic surface) than on PAA (abiotic surface).

In Fig2b the Δ_{adh} cells elongate symmetrically, which implies low and symmetric adhesiveness. Despite this, the forces that Δ_{adh} generate are quite close to WT. This seems counter intuitive.

Force at adhesion foci and asymmetry do not emerge from the same microscopic origins. As mentioned before, the force at adhesion foci, F_{foci} , is set by both the rupture force and the number of adhesive links, while the asymmetry in adhesion A_{cell} is set by the sub-cellular localization of the adhesive links, which can depend on many other cellular factors. Hence, A_{cell} and F_{foci} are not expected to be correlated. To know which of these two terms dominates in establishing the shape of the microcolony, we compared the different mutant strains. We have added a paragraph in the conclusion to clarify this point (Lines 222 to 232).

4. *Figure 4 i. 4b: Do the authors think that the elastic link to the surface extends the full length of the width of cells? In other words, for cells that get pushed into the 3D, does the link to the surface remain?*

Experiments in the literature have shown that the end-to-end distance L_0 of polymeric link can be rather large compared to cell size (Forero *et al.*, *PLoS Biol* 2006, Jacquot *et al.*, *J. Bio. Nano.* 2014). In our model, we used the experimental values reported by Forero *et al.* (see supplementary table 3). At the rupture force F_{link} used in our model, the size of the links is larger than cell width when they break. Therefore, we included in the calculation of $N_{2D/3D}$ the mechanical work W required to extend those links in the vertical dimension at second layer formation (see Eq.16).

5. *Supplementary Figures*
i. Supp Fig 1: Light yellow colors are very hard to read
ii. Supp. Fig 2b: the label for the error bars is missing
iii. line 305 refers to Suppl Fig 3B, but should be 4B.
iv. Supp Fig 7 caption: there is no "3G"

We have changed colors in Supp Fig1, added the description of error bars in Supp Fig2b and corrected references to figures in the Methods (Line 344) and in Supp Fig7.

6. *On lines 6-7 in the abstract the authors make a very general statement that all rod shaped bacteria have higher adhesiveness at the old poles. In my opinion, to avoid confusion they should clarify this point by first referencing the bacteria that they use and then generalizing to all real and modeled rod-shaped bacteria with the type of asymmetric adhesiveness that E. coli and P. aeruginosa have.*

We have made changes in the abstract in order to explicitly mention the strains used in the study.

7. *The introduction is quite short, too general, and a bit unclear to adequately provide context, introduce, motivate, and review the literature sufficiently for the interesting problem they address. For example, on line 16, what does "organized structures" mean? What "environmental conditions" are being referred to? Why does bacterial elongation compete with surface attachment? Please clarify these points.*

We have extended the introduction (Lines 15 to 50) in order to discuss the different points raised by

the reviewer.

8. *On lines 207-211 the authors introduce a weak argument for the potential ecological advantages that rod shaped bacteria have over coccoids in surface colonization. There are far too many assumptions built into this statement for it to be meaningful. Too many factors come into play during surface colonization to simply say that asymmetric adhesion confers an advantage. For example, if a coccoidal species had a shorter doubling time or strong cell-to-cell adhesion, one could imagine a similar surface colonization program but with the coccoid have the advantage. The authors should provide more context and support for this statement.*

We have removed this point. Our initial thought was motivated by the fact that if we consider two species with the same growth rate and the same force at adhesion foci, the tiling of spheres follows an hexagonal motif that is always more compact than the disordered tiling of cylinders. So for two equivalent amounts of biomass, the surface covered by the microcolony, which includes gaps, would be larger for rod-shaped bacteria than coccoids.

9. *Minor comments on awkward phrasing, grammar,*
a) *lines 186-191: The wording in these lines could be reworded to be more clear*

We have reformulated this phrase and moved it into the first paragraph of the discussion (Lines 228 to 232).

- b) *line 202: i think the authors mean "adhesins"*

We have replaced "adhesions" by "adhesive molecules" (Line 243).